# Sugar Transporters in *Plasmodiophora brassicae*: Genome-Wide Identification and Functional Verification

**DOI:** 10.3390/ijms23095264

**Published:** 2022-05-09

**Authors:** Liyan Kong, Xiaonan Li, Zongxiang Zhan, Zhongyun Piao

**Affiliations:** Molecular Biology of Vegetable Laboratory, College of Horticulture, Shenyang Agricultural University, Shenyang 110866, China; kly@stu.syau.edu.cn (L.K.); gracesleexn83@syau.edu.cn (X.L.)

**Keywords:** *Plasmodiophora brassicae*, sugar transporter, yeast functional complementation

## Abstract

*Plasmodiophora brassicae*, an obligate intracellular pathogen, can hijack the host’s carbohydrates for survival. When the host plant is infected by *P. brassicae*, a large amount of soluble sugar accumulates in the roots, especially glucose, which probably facilitates the development of this pathogen. Although a complete glycolytic and tricarboxylic acid cycle (TCA) cycle existed in *P. brassicae*, very little information about the hexose transport system has been reported. In this study, we screened 17 putative sugar transporters based on information about their typical domains. The structure of these transporters showed a lot of variation compared with that of other organisms, especially the number of transmembrane helices (TMHs). Phylogenetic analysis indicated that these sugar transporters were far from the evolutionary relationship of other organisms and were unique in *P. brassicae*. The hexose transport activity assay indicated that eight transporters transported glucose or fructose and could restore the growth of yeast strain EBY.VW4000, which was deficient in hexose transport. The expression level of these glucose transporters was significantly upregulated at the late inoculation time when resting spores and galls were developing and a large amount of energy was needed. Our study provides new insights into the mechanism of *P. brassicae* survival in host cells by hijacking and utilizing the carbohydrates of the host.

## 1. Introduction

The uptake of sugar across the boundary membrane is a primary event in the nutrition of all eukaryotic cells [1]. The regulation of sugar entry is a common biological strategy for modulating cellular activities. Increasing evidence shows that, during plant–pathogen co-evolution, microorganisms have evolved sophisticated mechanisms to hijack sugar fluxes from their hosts, especially for obligate biotrophs [2,3]. *Plasmodiophora brassicae*, an intracellular obligate biotroph, is dependent on the nutrients of its host and establishes a long-term feeding relationship by stimulating the early steps of phloem differentiation and phloem-specific expression of sugar transporters within developing galls [4]. Inspection of the *P. brassicae* genome showed a complete glycolytic and tricarboxylic acid cycle (TCA) cycle, indicating a capacity for the utilization of carbohydrates from the host for energy production. In addition, a unique set of potential sugar transporters has been predicted in the *P. brassicae* genome [5]. Hence, *P. brassicae* may possess a sugar transporter system.

*Plasmodiophora brassicae* is still disputed over its classification status, which is currently generally considered a protist [6]. Along with plant pathogens of the genera *Polymyxa* and *Spongospora*, *Plasmodiophora* comprises the order Plasmodiophorida, which belongs to the class Phytomyxea (plasmodiophorids) [7,8]. Currently, this class is considered part of the protist supergroup Rhizaria and has been placed within the phylum Cercozoa and the subphylum Endomyxa [9,10]. *Plasmodiophora*
*brassicae* has a complex life cycle, generally comprising three stages: survival in the soil, primary infection of root hairs, and cortical secondary infection. Members of the Brassicaceae family are thought to be potential hosts for *P. brassicae*, parasitizing the host root cell and covering it with a cystic structure to facilitate nutrition exchange between *P. brassicae* and its host.

Sugar is the main product of photosynthesis, it regulates plant metabolism and acts as a metabolic substrate and signaling molecule [11,12]. Sucrose, as the final product of photosynthesis, is transported from the source organ to the sink through the sieve element/companion cell complex of the phloem. Sink organs in higher plants include developing seeds, fruit, and roots. In addition, infection by biotrophic microbes could establish an extra sink, causing reprogramming and carbon partitioning in the host [13,14]. Sugar metabolism plays a pivotal role in governing various kinds of plant–pathogen interactions [15]. The obligate biotrophic fungus *Puccinia striiformis f*. sp. *tritici (Pst)* colonizes the mesophyll cells and the apoplast of wheat leaves to obtain carbohydrates for energy and carbon [14]. The fungal pathogen *Ustilago maydis* causes changes in carbohydrate metabolism in the infected maize leaves [16]. Notably, changes in carbohydrate metabolism in maize contribute to altered *U. maydis* susceptibility [16]. The pathogenicity of some soil-borne pathogens, such as *Verticillium dahlia*, *Fusarium oxysporum*, *Phytophthora infestan*, and *Rhizoctonia solani*, was affected by changes in the sugar metabolism of the host plant [17,18,19,20]. In terms of *P. brassicae*, several studies have reported that soluble sugars (hexose and sucrose) and starch accumulate in the galls of *P. brassicae*-infected plants, and the development of galls is associated with altered carbohydrate metabolism and partitioning in host plants [21,22]. Sugar efflux to the pathogen could be a carbon supply strategy for pathogen reproduction [23]. Furthermore, exogenous carbon application at the infection site enhances the virulence of pathogens [16]. Sugar uptake and release from cellular exchanges rely on sugar transporters [24].

Sugar transporters, including three common sugar transporter families, namely sucrose transporters (SUTs) [25], monosaccharide transporters (MSTs) [26], and sugar, will eventually be exported transporters (SWEETs) [2]. MSTs and SUTs are members of the major facilitator superfamily (MFS) with 12-transmembrane helices that function in monosaccharide and sucrose sugar influx, respectively [27,28]. SWEETs function as uniporters, facilitating the diffusion of sugars across cell membranes and participating in phloem transport [2,29]. SWEET belongs to MtN3_slv or the PQ-loop family that contains seven transmembrane helices (TMHs) in eukaryotes and three TMHs in prokaryotes. SWEETs are widely found in various organisms, which have been identified not only in plants but also in fungi, as well as in bacteria and animals, including humans. Animals contain only 1 gene coding SWEET, which is encoded by *SLC50A1* gene, excluding *Caenorhabditis elegans*, which contains 7 genes [30,31,32,33]. Currently, the most comprehensive study of sugar transport in eukaryotic fungal systems has been performed on *Saccharomyces cerevisiae*. A total of 20 genes encoding hexose transporters belonging to the MFS superfamily have been identified in *S. cerevisiae*, and 18 genes including *Hxt1p-Hxt17p* and *Gal2p* function as the transport of glucose, fructose, and mannose. Snf3p and Rgt2p act as glucose sensors to regulate the expression of hexose transporters [34,35,36]. Additionally, in biotrophic plant-interacting fungi, a few sugar transporters have been characterized, such as the MSTs (*AmMst1*, *AmMst2*) from *Amanita muscaria* [37], two high-affinity hexose transporters (*Tmelhxt1*, *Tmel2281*) from *Tuber melanosporum* [38], and 15 putative sugar transporters in the genome of the *Laccaria bicolor*, which function in carbon assimilation [39]. Unlike the ectomycorrhiza formed by some fungi, *P. brassicae* obligately parasitizes the roots of host plants, and plants transport photosynthetic products to the roots, and these products are hijacked by *P. brassicae*. Thus, it is unclear whether *P. brassicae* has the same hexose transport system as other fungi.

Thus far, our knowledge of sugar transporters in *P. brassicae* is limited. To bridge this gap, we used data obtained from the sequenced genome of *P. brassicae* in the National Center for Biotechnology Information (NCBI) database [5,6,40,41] to search for potential hexose sugar transporters and used the *hxt-null S. cerevisiae* strain EBYVW.4000 [42] to verify the function of hexose sugar transport.

## 2. Results

### 2.1. Soluble Sugar Content in Cabbage Roots after Infection with P. brassicae

To determine if the clubroot formation affects soluble sugar content through the carbohydrate metabolism of the host plants, the soluble sugar content (glucose, fructose, and sucrose) of the roots was analyzed at four different stages after infection (from the second week to the fifth week). Among these, sucrose is the predominant soluble sugar in the roots, followed by glucose and fructose. From the second to the fourth week after inoculation, the sucrose and glucose content did not change significantly, while the fructose content gradually increased. In addition, the sucrose and fructose content was not affected between the infected plants and healthy plants. However, the glucose content increased greatly in the roots of infected plants, which was significantly higher than that of non-infected plants at 5 weeks post-inoculation (Figure 1).

### 2.2. Identification of Putative Sugar Transporters of P. brassicae

A total of 17 putative sugar transporters were identified from the *P. brassicae* genome through a BLAST search and Pfam domain analysis. The selected proteins containing the sugar transporter signature belong to the Sugar_tr (PF00083), MtN3_slv (PF03083), or PQ-loop (PF04193) domains (Table 1). General information about the 17 proteins is summarized in Table 1. Among these proteins, the coding sequence (CDS) length ranged from 690 to 5379 bp, and the length of the protein sequence ranged from 229 to 1792 amino acids, with predicted molecular weights ranging from 25.31 to 159.50 kDa. The theoretical isoelectric point (pI) ranged from 5.67 to 9.97. These proteins were classified into two sub-families: 4 proteins containing the *Sugar_tr* domains belonging to the MFS superfamily, and 13 proteins containing the MtN3_slv or PQ-loop domains are members of the SWEET superfamily. The CDS length of genes in the MFS superfamily is generally larger than that of the genes of the SWEET superfamily, which may be related to the number of TMHs. Among the 17 putative sugar transporter genes, 14 genes were successfully amplified from the resting spores of *P. brassicae*, except for PQ-10, PbHxt1, and PbHxt2 (Appendix A).

### 2.3. Transmembrane Helices and Motifs of Sugar Transporters in P. brassicae

SUTs and MSTs containing the “Sugar_tr” domain (PF00083) belong to the MFS, which usually contains 12 TMHs [43]. SWEETs, another family of sugar transporters, are characterized by seven TMHs and the MtN3_slv or PQ-loop domain in eukaryotes [33]. The TMHs of the putative sugar transporters in *P. brassicae* showed diversity. For the putative transporters with “Sugar-tr” domains, PbHxt1 and PbHxt4 had 10 TMHs, PbHxt2 had 11 TMHs, and PbHxt3 had 12 TMHs, as predicted in other organisms. For the remaining 13 proteins with MtN3_slv or PQ-loop domains, most contained seven and six TMHs, whereas three proteins (PQ-8, PQ-10, PQ-13) contained only four or five TMHs (Figure 2a).

To further investigate the structural diversity, the protein sequences of *P. brassicae* sugar transporters were submitted to the online MEME program for analysis of the conserved motifs, and the motif sequences and annotations were further predicted by Pfam. Proteins with “Sugar_tr”, “MtN3_slv”, or “PQ-loop” domains were selected for motif analysis. The four proteins with the Sugar_tr domain contained 4–5 conserved motifs, the length of which ranged from 20 to 38 amino acids, and the Sugar_tr domain was predicted in motifs 1, 2, 3, and 4. Furthermore, three motifs (motifs 1, 2, and 3) exhibited high conservation. The protein motif distributions of Pbhxt3 and Pbhxt4 were similar, both of which contained motif 1 in the N-terminal region and motifs 3, 2, and 4 in the C-terminal region (Figure 2b). Five conserved motifs were found for 13 proteins with MtN3_slv or PQ-loop domains. The length of the conserved motifs ranged from 27 to 50 amino acids, and the PQ-loop domain was predicted in motifs 1, 2, 3, and 5. Five proteins (PQ-1 to PQ-5) shared the same motif distribution, where motifs 1, 4, and 5 were in the N-terminal region and motifs 2 and 3 were in the C-terminal region. Four proteins (PQ-6, PQ-8, PQ-11, and PQ-13) possessed two conserved motifs (1 and 2), and PQ-7 contained only motif 1. Strangely, we found three proteins (PQ-9, PQ-10, and PQ-12) with no conserved motif (Figure 2c). PQ-9 and PQ-12 harbored the MtN3_slv domains. Thus, an additional motif analysis was carried out on PQ-9 and PQ-12. We found that both proteins had five conserved motifs, and MtN3_slv domains were annotated as conserved motifs 1 through 5 (Appendix A).

### 2.4. Phylogenetic Analyses of Hexose Transporters Identified in P. brassicae

To better explore the phylogenetic relationship of the sugar transporters identified in *P. brassicae*, all sugar transporter proteins were subjected to an extensive BLAST search in NCBI’s nr database to determine their homologous sequences in other organisms, based on the criterion that sequence identity > 40%, *E* value < 1 × 10^−20^, and query coverage > 60% without redundant or uncharted proteins. However, these *P. brassicae* sugar transporters’ protein sequences showed low identity with other organisms’ protein sequences (sequence identity about 50%). Therefore, only 36 protein sequences were employed to construct the phylogenetic tree (Figure 3). These homologous proteins were divided into four clades. Ten PQ-loop proteins (PQ-1, PQ-2, PQ-3, PQ-4, PQ-5, PQ-6, PQ-8, PQ-10, PQ-11, and PQ-13) were clustered in the first group. In this group, PQ-2 and PQ-6 clustered with the PQ loop repeat proteins of *Pelomyxa schiedti* (KAH3767290.1) and *Leiotrametes lactinea* (KAH9893501.1), respectively. PQ-10 clustered with the cleft lip and palate transmembrane protein of *Phoenix dactylifera* (XP 008786461.2) and the CLPTM1-like membrane protein of *Arachis ipaensis* (XP 016169702.1). In the second group, PQ-7 clustered with the lysosomal Cystine transporter of *Spizellomyces punctatus* (XP 016608459.1). Interestingly, PQ-9 and PQ-12 were in the third group, without any homologous proteins from other organisms. All MFS family proteins of *P. brassicae*, including PbHxt1, PbHxt2, PbHxt3, and PbHxt4, were grouped into the fourth group. PbHxt1 and PbHxt2 were homologous with other MFS transporters of *Chloroflexi bacterium* (RPJ49763.1) and *Anaerolineae bacterium* (MBI4926876.1). PbHXt4 clustered with some substrate transporters and hexose transporters of fungi or water bears (*Hypsibius exemplaris*, OQV25459.1). Based on the above analysis, only PbHXt4 clustered with a characterized hexose transporter and sugar transporter (*Planoprotostelium fungivorum*, PRP86522.1). Considering the lower identity of homologous sequences, these sugar transporters of *P. brassicae* were probably unique, and sugar transporter activity requires further validation.

### 2.5. Hexose Transport Activity of P. brassicae Sugar Transporters in Yeast

To verify the functionality of the *P. brassicae* sugar transporters, hexose uptake was evaluated through the heterologous expression of these putative transporters in the hexose transport-deficient yeast strain EBY.VW4000. As shown in Figure 4, eight transporters (PbHxt3, PQ-3, PQ-4, PQ-5, PQ-7, PQ-9, PQ-12, and PQ-13) could restore EBY.VW4000 growth on 2% glucose. Eight transporters (PQ-2, -4, -5, -7, -9, -11, -12, and -13) had fructose transport activity. Among these, five proteins could transport fructose and glucose (PQ-4, -5, -7, -9, and -12). Only PbHxt3 with the sugar_tr domain had the ability to transport glucose, but the proteins with the PQ-loop or MtN3_slv domains had dual transportability of fructose and glucose. In particular, PQ-5 and PQ-7 had stronger transport activity with both glucose and fructose than the other transporters. In addition, growth through yeast spots indicated that these sugar transporters preferentially transported glucose over fructose (Figure 4).

### 2.6. Expression Patterns of Glucose Transporters in P. brassicae

Among these putative sugar transporters, eight proteins functioned in glucose transport. Considering the significant accumulation of glucose content in the roots at the late inoculation stage, quantitative real-time PCR was performed to analyze the expression characteristics of these glucose transporters in the roots at different times of inoculation. The expression of these transporters was strongly induced 4 weeks after inoculation. The expression of PQ-3 and PQ-5 at 5 weeks post-inoculation was upregulated 60-fold compared to that at the third week. Interestingly, the expression of these transporters in the spike coincided with the increase in soluble sugar in roots infected by *P. brassicae*. Six of eight glucose transporters, namely PQ-3, PQ-4, PQ-5, PQ-7, PQ-9, and PQ-13, were increased at 4 weeks post-inoculation but were still significantly higher than that at 3 weeks post-inoculation. Only the expression of PbHxt3 and PQ-12 showed no significant differences in the fourth week. These results indicate that sugar transport activity was positively correlated with the expression of related genes during the formation of galls (Figure 5).

## 3. Discussion

Sugar transporters play a key role in the ability of obligate parasites to hijack the host’s carbohydrates for survival. Here, we screened 17 putative transporters from the genome of *P. brassicae*. The structure of these transporters showed variations compared with other organisms, especially in the number of TMHs. Yeast functional complementation experiments indicated that eight transporters had the ability to transfer hexose, including glucose and fructose. The expression of these transporters coincided with the increase in soluble sugar in roots infected with *P. brassicae*.

*Plasmodiophora brassicae* is an obligate parasite that parasitizes the roots of cruciferous plants. Although our knowledge of its life cycle through microscopic observation technology has grown considerably in recent years, its study remains limited, because it cannot be isolated and cultured in vitro [7,44]. The pathogen attempts to manipulate the carbohydrate metabolism of the host in its favor, and it has been reported that the sink-specific genes encoding cell wall invertase in host plants are induced during pathogen infection. Cell wall transformation hydrolyzes sucrose into glucose and fructose and then introduces hexose into the target cells [45,46]. Transcriptomic analysis of *A. thaliana* and *P. brassicae* showed that sucrose synthase (SUS) and sugar permeases were induced during gall formation of *P. brassicae*, and cytoplasmic invertase mutant cinv1,2 had resistance to clubroot disease at 26 days post-inoculation (DPI), indicating that *P. brassicae* formed a new sink [47]. Soluble sugar accumulates in large amounts in infected organs during pathogen infection, and it has been proposed that sucrose and hexose accumulation in the apoplast is caused by enhanced sugar efflux from host cells [2,48]. *Arabidopsis thaliana sweet11*, *12* double mutants exhibited delayed *P. brassicae* development due to impaired sugar transport activity toward the pathogen [4]. In this study, we observed changes in soluble sugar content in infected Brassica roots, mainly in glucose; similar results were also observed in *P. brassicae*-infected *Brassica rapa* [21]. At the fifth week after inoculation, the sucrose content of the infected plants was lower than that of the healthy plants, but the glucose content was significantly higher than that of the healthy plants. This indicates that more sucrose was converted to glucose in the roots of infected plants. Soluble sugars provide a carbon source for the growth of *P. brassicae*, and glucose is preferentially utilized. This view has been shown in previous studies to favor hexose over sucrose uptake by fungal biotrophs [49,50,51,52,53].

Efficient transport systems are necessary for *P. brassicae* to take up carbon nutrients, but this mechanism is unclear. To elucidate these mechanisms in *P. brassicae*, we took advantage of the genome sequencing data of *P. brassicae* in the NCBI database to identify sugar transporters. A total of 17 potential sugar transporter genes were identified according to the annotations. Four proteins contained *sugar_tr* (PF00083) domains belonging to the MFS. MFS transporters usually contain 12 TMHs [54,55], while only one protein contained 12 TMHs; the other proteins contained 10 or 11 TMHs. Sugar transporters with an incomplete structure have been reported in *Trichoderma reesei* [56]. Previous studies have shown that eukaryotic SWEETs contain seven TMHs, while prokaryotic semiSWEETs contain only three TMHs, with a set of three TMHs constituting one MtN3 or PQ-loop unit [29,33,57]. In addition, an extraSWEET protein consisting of 14 TMHs has been reported in *Vitis vinifera* [58], and 15 extraSWEET and 25 superSWEET TMHs were also identified in wild rice and oomycetes, respectively [59]. In our study, 11 proteins containing PQ-loop domains and 2 proteins with *MtN3/saliva* domains were identified. Thus, these proteins may encode SWEET transporters in *P. brassicae*. Our analyses revealed that PQ-loop transporters of *P. brassicae* contain a diverse number of TMHs (4–7; Figure 2), and nearly half of them possess seven TMHs. Analysis of seven TMH proteins suggested that they all contained two MtN3 or PQ-loop units. Interestingly, only four TMHs were found in the protein sequence of PQ-13, 3-TMH structure, which was previously only reported in bacteria and archaea [60]. The semiSWEETs of *Bradyrhizobium japonicum* may mediate sucrose transport [57]. Thus, we speculated that the protein PQ-13 may encode a semiSWEET in *P. brassicae*. At present, there are two main explanations for the evolutionary mechanism of SWEETs. Xuan et al. [57] suggested that eukaryotic SWEETs were generated by the duplication and fusion of semiSWEETs, while Hu et al. [61] believed that SWEETs were generated by the fusion of archaeal and bacterial semiSWEET. Our study showed that PQ-loop transporters with 4, 5, 6, and 7 TMHs are present in the *P. brassicae* genome, implying that duplication and fusion of SWEETs may be ongoing in the genome. We found that 14 putative sugar transporters could be cloned from resting spores, indicating that these genes may be expressed in a specific period (Appendix A).

Many microbial sugar transporters have been identified and functionally validated, such as in *Saccharomyces cerevisiae* [34], *Colletotrichum gloeosporioides* [62], ectomycorrhizal fungus *Tuber melanosporum* [38], *Bradyrhizobium*
*japonicum* [57], arbuscular mycorrhiza fungus [63], and *Pseudomonas stutzeri* A1501 [64]. Nevertheless, *P. brassicae* is not a fungus or bacteria, but is an obligate parasite within the class Phytomyxea (plasmodiophorids) of the protist supergroup Rhizaria [7]. We performed a BLASTp search in the NCBI database using the putative sugar transporter sequences of *P. brassicae* to identify highly homologous sugar transporters from other species. However, most of the proteins of *P. brassicae* failed to cluster with known sugar transporters from other species. In addition, the alignment results of the protein sequences showed very low similarity. This suggests that these proteins are unique to the *P. brassicae* genome.

Hexose-deficient yeast EBY.VW4000 has been used in many studies to verify whether the sugar transporter has hexose transport activity [57,65]. In this study, we used this method to identify some *P. brassicae* sugar transporters with glucose and fructose transport activities and to determine whether they have sucrose transport activity. We next plan to further validate the function of these *P. brassicae* sugar transporters using a Förster resonance energy transfer (FRET) sensor-based screen [29]. The relative real-time fluorescence quantitative results showed that the expression levels of these glucose transporters were consistent with the results of the significant accumulation of glucose in the roots. We hypothesized that the host plant sucrose was hydrolyzed to glucose and that glucose was transported to the resting spores by these *P. brassicae* glucose transporters. Previous studies have found that a soluble disaccharide, trehalose, which is synthesized from glucose as a substrate by the trehalose-6-phosphate synthase gene (TPS), is abundant in the resting spores of Brassica plants, and the accumulation pattern of trehalose correlated with the *PbTPS1* gene from *P. brassicae* [5,66]. The glucose transported by the glucose transporter of *P. brassicae* may be used to synthesize trehalose, which needs to be verified in the future.

In conclusion, a large amount of glucose was accumulated in the roots of *Brassica rapa* during clubroot development, and 17 potential sugar transporters were identified. Eight upregulated candidate sugar transporters transported glucose with the development of *P. brassicae*. Based on the above results, we propose that *P. brassicae* may possess a transporter system to transport host plant sugars from the apoplast into the cytoplasm through its own sugar transporters.

## 4. Materials and Methods

### 4.1. Plant Material and P. brassicae Inoculation

The plant material used as a host was the Chinese cabbage susceptible variety ‘*BJN3-2*’. The plants were maintained in the culture room under a 16 h light/8 h dark photoperiod at 25 °C. A single-spore isolate (Pb4) of *P. brassicae* was applied. The resting spores were collected from homogenized clubbed roots and diluted to a density of 1 × 10^7^ spores/mL with sterile distilled water until inoculation. The 2-week-old seedlings of Chinese cabbage were inoculated with *P. brassicae* by injection, and 1 mL of a resting spore suspension was injected into the soil near the hypocotyl for each plant. The roots of cabbage were sampled at 1, 2, 3, 4, and 5 weeks after inoculation and used for the determination of soluble sugar content and gene expression analysis. Non-inoculated plants were used as the control. Each treatment was carried out with three biological replicates, and each replicate contained six plants. This experiment was carried out in the greenhouse of Shenyang Agricultural University in 2021.

### 4.2. Soluble Sugar Extraction and Content Determination

Soluble sugars were extracted from the roots of Chinese cabbage sampled at 2, 3, 4, and 5 weeks after *P. brassicae* infection. The soluble sugar was extracted using a method reported previously [21]. The column DB-5 MS (Agilent, J&W Scientific, Folsom, CA, USA, 30 m × 0.25 mm × 0.25 μm) was selected for gas quality analyses. The inlet temperature was 300 °C, the split ratio was 10:1, the carrier gas was high-purity helium, and the flow rate was 1 mL/min. The heating program was as follows: 120 °C for 3 min, 5 °C/min to 210 °C for 5 min, and 15 °C/min to 300 °C for 10 min. The ion source temperature was 220 °C, the interface temperature was 280 °C, the solvent removal time was 3 min, and the scanning m/z was 45–500. Soluble sugars were extracted from three biological replicates at each time point, as in the aforementioned treatments. Gas quality analyses were repeated three times for each treatment.

### 4.3. Total RNA Extraction and Quantitative Real-Time PCR

The total RNA was extracted from the root using TRIZOL reagent (Tiangen, Beijing, China), following the manufacturer’s guidance. Reverse transcription was performed using a Thermo Scientific ReverAid Kit. Quantitative real-time RT-PCR was performed using the SYBR Green Premix system (MonAmp™ ChemoHS qPCR Mix Monad, Shanghai, China). PCR reactions were carried out in triplicate with three independent RNA samples, and the primers are listed in Appendix A. The 2^−ΔΔCt^ [67] method was used to analyze the relative gene expression level. Statistical analyses were performed with a one-way analysis of variance using direct statistical software.

### 4.4. Screening Putative Sugar Transporters of P. brassicae

Putative sugar transporters were screened according to the keywords “*sugar-tr*”, “*MtN3_slv*”, or “*PQ-loop*” from the whole genome sequencing data of *P. brassicae* in the National Center for Biotechnology Information database (NCBI) (https://www.ncbi.nlm.nih.gov/, accessed on 3 January 2021) [5,40,41,68].

Proteins including Sugar_tr (PF00083), MtN3_slv(PF03083), and PQ-loop (PF04193) domains were selected and used for further domain verification by the Pfam database (http://pfam.xfam.org/, accessed on 14 April 2022). Redundant sequences and fragment sequences were removed. Proteins with “*sugar-tr*” domains were named Pbhxt1–Pbhxt4, and those with PQ-loop or MtN3_slv domains were named PQ-1 to PQ-13. The protein molecular weight (kDa) and isoelectric points (pI) were calculated according to the online ExPASy2 database (https://www.expasy.org/resources/compute-pi-mw, accessed on 14 April 2022).

### 4.5. Transmembrane Helices and Motifs of P. brassicae Putative Sugar Transporters

The online tool TMHMM Server 2.0 https://services.healthtech.dtu.dk/service.php?TMHMM-2.0 (accessed on 14 April 2022) was utilized to forecast the protein structure of sugar transporters in *P. brassicae*. The online MEME tool (https://meme-suite.org/meme/tools/meme, accessed on 14 April 2022) was used to identify conserved protein motifs of putative sugar transporters [67]. MEME was employed using the following parameters: optimum width: 20–50; site distribution: zero or one occurrence per sequence; maximum number of motifs: 5; and other settings followed the default options.

### 4.6. Clustering Analysis of P. brassicae Glucose Transporters

Each protein sequence was subjected to protein BLAST in NCBI’s nr (Non-Redundant Protein Sequence Database). Proteins from other organisms having a BLASTP hit (percent identity > 40%; *E* value < 1 × 10^−20^; query coverage, 60%) were selected to construct the phylogenetic tree [69]. All selected protein sequences were in the Appendix A. Phylogenetic analysis was performed with all protein sequences. Multiple sequence alignment was carried out using *Clastal*W (https://www.ebi.ac.uk/Tools/msa/clustalw2/, accessed on 14 April 2022), the unrooted phylogenetic tree was constructed by MEGA6 with a bootstrap of 1000 replicates using the neighbor-joining (NJ) method, and Poisson correction was chosen as the distance parameter.

### 4.7. Vector Construction and Functional Complementarity of Hexose-Deficient Yeast EBY.VW4000

The yeast expression vector pDR195 was applied to the analysis of the hexose transport function. The CDSs of these sugar transporter genes with *XhoI* and *BamHI* restriction sites were cloned into the yeast expression vector pDR195 using a TaKaRa In-Fusion**^®^** HD Cloning Kit (Dalian, China, catalog number 639650), following the manufacturer’s instructions. AtSWEET1 was used as a positive control for glucose uptake, and AtSWEET16 was used as a positive control for fructose uptake. Empty vector pDR195 was used as a negative control. The constructed expression vector was transferred to the hexose transport-deficient yeast strain EBY.VW4000, which cannot grow on hexose medium due to concurrent knockout of 20 endogenous transporter genes, but can grow on maltose medium. Transformants of the EBY.VW4000 strain were grown on liquid synthetic deficient (SD)/-uracil media supplemented with 2% maltose. For complementation growth assays, the concentration of EBY.VW4000 yeast cell suspension was adjusted to OD600 = 0.5. Serial dilutions (10^−1^, 10^−2^, 10^−3^, and 10^−4^) were plated dropwise on solid SD/-uracil medium consisting of 2% maltose and 2% glucose/fructose. These plates were grown at 30 °C for 3 days and used for imaging.

### 4.8. Statistical Analysis

The data on sugar content are presented as the means ± SEs (standard errors). Other statistical evaluations and significance tests were performed via Student’s *t*-tests at * *p* < 0.01 or ** *p* < 0.05 with the SPSS statistical software (Version 19.0; SPSS, Inc., Chicago, IL, USA). The data were graphically analyzed using GraphPad Prism V8.0.2 (San Diego, CA, USA).

## Figures and Tables

**Figure 1 ijms-23-05264-f001:**
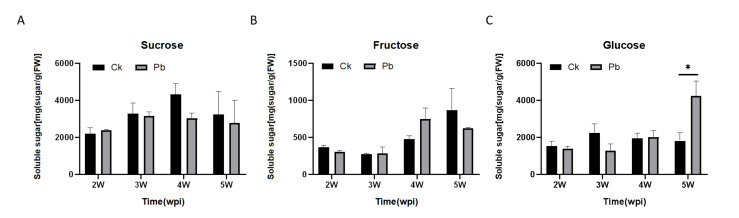
Sugar content in the roots of Chinese cabbage after *Plasmodiophora brassicae* infection. (**A**), Sucrose, (**B**), Fructose, (**C**), Glucose. Ck, inoculation with distilled water; Pb, inoculation with *P. brassicae*. FW, Fresh weight. The horizontal axis represents the time point from 2 to 5 weeks post-inoculation (wpi), and the ordinate represents the value of the sugar content. The results are means ± SEs from three independent biological repeats. The asterisks indicate *p*-values (* *p* < 0.05) according to Student’s *t*-test. The horizontal axis is the time point from 2 to 5 weeks post-inoculation (wpi) independent biological repeats.

**Figure 2 ijms-23-05264-f002:**
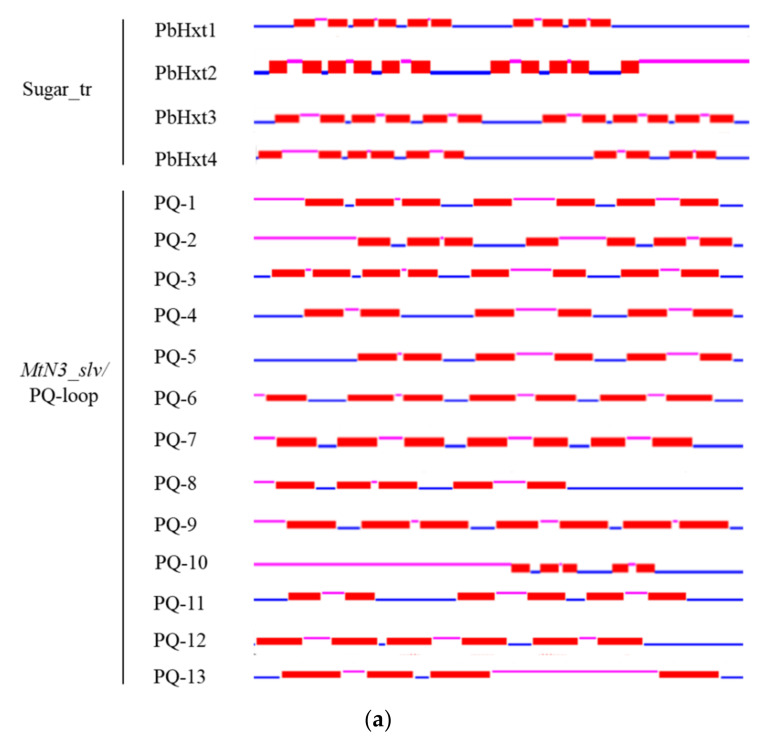
Transmembrane domain structures and distribution of conserved motifs of the sugar transporters in *Plasmodiophora brassicae*. (**a**) Predicted transmembrane domain structures using TMHMM Server 2.0. Blue lines indicate regions with cytosolic locations, pink lines indicate regions with apoplastic localization, and red boxes indicate putative transmembrane domains. (**b**) Distribution of protein conserved motifs with Sugar-tr domains. (**c**) Distribution of protein-conserved motifs with PQ-loop domains. Conserved motif analysis was performed using MEME 2.0 software, as described in the methods. Each motif is indicated by a colored box numbered from 1 to 5. The lengths of the motifs in each protein were proportional.

**Figure 3 ijms-23-05264-f003:**
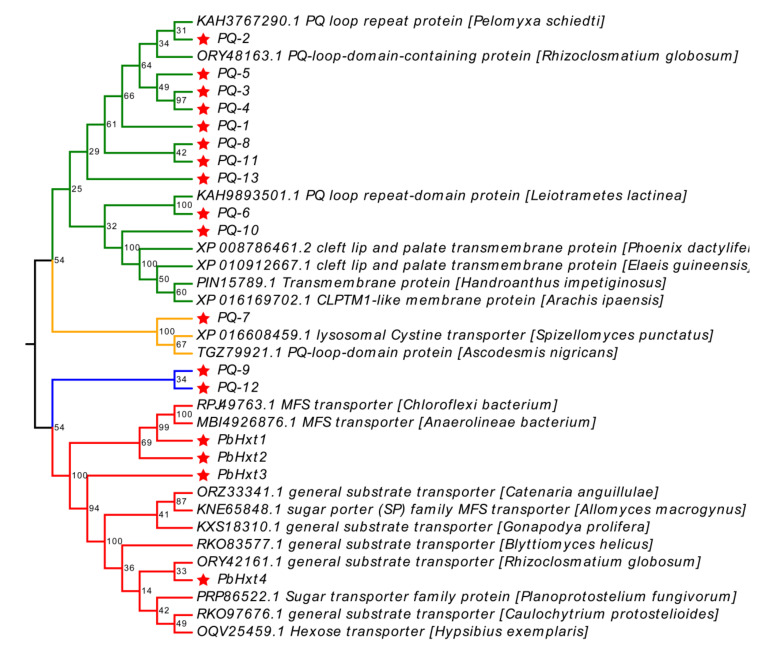
Phylogenetic relationships of Plasmodiophora brassicae sugar transporters. The unrooted phylogenic tree was constructed using 36 amino acid sequences. The phylogenic tree was constructed with MEGA6 using the neighbor-joining method. The red stars represent the putative sugar transporters of P. brassicae. Each colored branch represents a different clade. The letters in front of the other sequences represent the GenBank number of the protein. The words in square brackets “[]” show the genus and species name.

**Figure 4 ijms-23-05264-f004:**
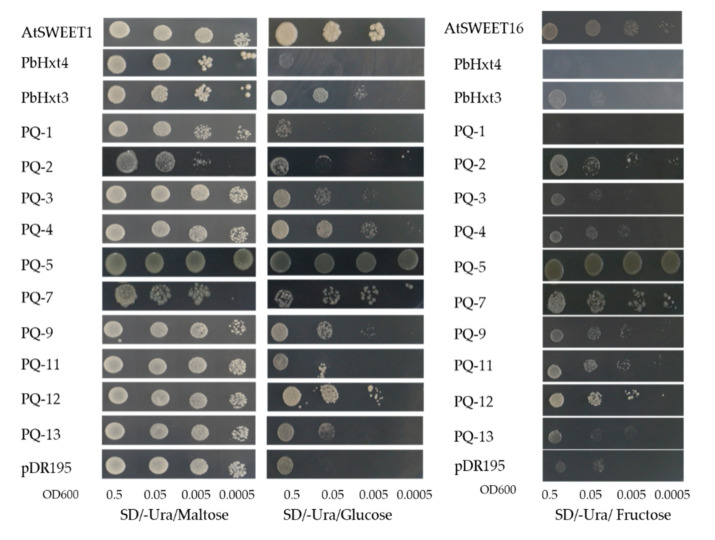
Complementation of yeast EBY.VW4000 with sugar transporter genes of *Plasmodiophora brassicae*. Growth of the yeast mutant strain EBY.VW4000 expressing the *P. brassicae* sugar transporter genes in SD (-Ura) solid media supplemented with 2% maltose, 2% glucose, and 2% fructose. To verify that the EBY.VW4000 strain grew on a non-hexose medium, 2% maltose was used as a positive control. AtSWEET1 was used as a positive control for glucose uptake. AtSWEET16 was used as a positive control for fructose uptake. The pDR195 empty vector was used as a negative control. Images show growth at 30 °C for 3 days.

**Figure 5 ijms-23-05264-f005:**
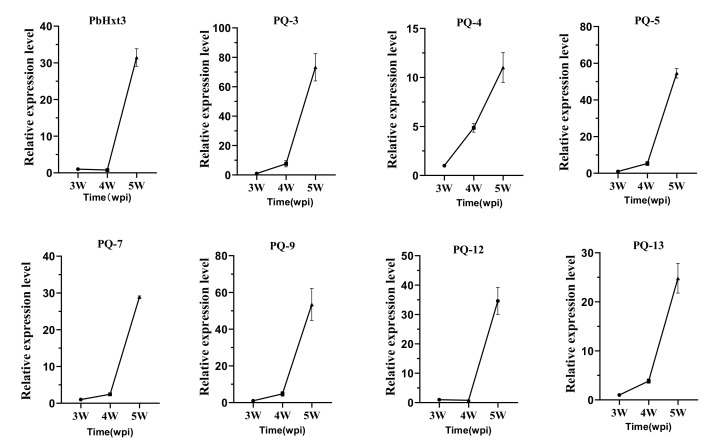
Relative expression level of glucose transporter genes in *P. brassicae*. qRT-PCR was used to quantify gene expression in cabbage roots at 3, 4, and 5 weeks post-inoculation (wpi) with *P. brassicae*. Transcript levels were normalized to the *Pbactin* gene. Three biological replicates were analyzed, and three technical repeats were performed per sample. Error bars indicate the standard errors between three biological repeats. Horizontal axis is time point from 3 to 5 weeks post-inoculation (wpi).

**Table 1 ijms-23-05264-t001:** Information on the putative sugar transporters of *Plasmodiophora brassicae*.

Gene Name	GenBank ID	Bp ^1^	Aa ^2^	MW ^3^ (kDa)	pI ^4^	Domain ^5^	Domain Position	TMH ^6^	Subfamily
PbHxt1	SPQ96265.1	1515	504	54.15	6.71	Sugar_tr	89–429	10	MFS superfamily
PbHxt2	SPQ93932.1	5379	1792	159.50	5.66	Sugar_tr	34–495	11	
PbHxt3	SPQ95410.1	1407	468	50.09	6.60	Sugar_tr	31–467	12	
PbHxt4	SPQ99469.1	1479	492	52.81	5.67	Sugar_tr	10–470	10	
PQ-1	SPQ94201.1	885	294	31.78	9.57	PQ-loop	35–93, 187–240	7	SWEET superfamily
PQ-2	SPQ94044.1	1041	346	38.05	9.23	PQ-loop	76–134, 252–308	7	
PQ-3	SPQ97464.1	879	292	31.43	8.36	PQ-loop	37–88, 184–237	8	
PQ-4	SPQ97733.1	864	287	31.97	8.50	PQ-loop	38–92, 182–238	6	
PQ-5	SPR00585.1	876	291	32.71	8.76	PQ-loop	37–93, 188–241	6	
PQ-6	CEO98791.1	729	242	26.62	8.60	PQ-loop	14–71, 144–200	7	
PQ-7	CEP03599.1	831	276	30.04	9.57	PQ-loop	12–63, 163–210	7	
PQ-8	SPQ93336.1	861	286	30.65	9.47	PQ-loop	17–73	5	
PQ-9	SPQ94026.1	690	229	25.31	9.62	MtN3_slv	19–100, 142–221	7	
PQ-10	SPQ96444.1	1824	607	69.30	6.8	PQ-loop	472–506	5	
PQ-11	SPQ97201.1	918	305	32.97	9.59	PQ-loop	24–80	6	
PQ-12	SPQ98447.1	736	245	26.01	8.81	MtN3_slv	139–219	6	
PQ-13	CEP03554.1	747	248	26.70	9.97	PQ-loop	42–99, 164–209	4	

^1^ The CDS Length. ^2^ The amino acid sequence length. ^3^ The molecular weight of protein. ^4^ The isoelectric point of the protein. ^5^ The domains were predicted by the pfam database. ^6^ The number of transmembrane helices was predicted using TMHMM Server v2.0.

## Data Availability

GenBank accession numbers are listed with the sequence data generated in this study.

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
