# Peer review of "Sugar Transporters in Plasmodiophora brassicae: Genome-Wide Identification and Functional Verification"

_ijms, 2022, doi:10.3390/ijms23095264_

Round 1

Reviewer 1 Report

Reviewed article is interesting, however needs few changes:

1. All Latin names, such as Plasmodiophora brassicae (lines 114, 189), P. brassicae (lines 149, 155, and other), Pelomyxa schiedti (207) and many other should be write in italic. 

2. Line 81. This sentence should be: ".... in plants but also in fungi, as well as in bacteria and animals, including humans"

3. Line 83 - Unclear. There are more than 20 genes belonging to the MFS superfamily. Please write more details.

4. SWEET was detected not only in plants, but also in animals and humans. It is encoded by SLC50A1 gene. Animals contain only 1 gene coding SWEET, excluding Caenorhabditis elegans which contains 7 genes. I think that this information should be included into text.

Reviewer 2 Report

The manuscript contains interesting research results that broaden the state of knowledge on a given topic. I propose to make some minor corrections.

Comments:

Citation: Lastname, F.; Lastname, F.; Lastname, F Title. - this fragment is not completed.

Line 369, where and in which year was the experiment performed?

Line 448, please provide full details of the manufacturer of the statistical software.

References, please delete the oldest publications, before 2010 (1-4, 8, 11, 17-18, 24, 26(!), 27, 30, 36, 39-41, 47, 51-53, 57, 63(!), 65-67, 69).  
